# Chromosome organization shapes replisome dynamics in *Caulobacter crescentus*

Chen Zhang [1], Asha Mary Joseph [2], Laurent Casini[3], Justine Collier [3], Anjana Badrinarayanan [2] & Suliana Manley [1] ✉

DNA replication in bacteria takes place on highly compacted chromosomes, where segregation, transcription, and repair must occur simultaneously. Within this dynamic environment, colocalization of sister replisomes has been observed in many bacterial species, driving the hypothesis that a physical linker may tether them together. However, replisome splitting has also been reported in many of the same species, leaving the principles behind replisome organization a long-standing puzzle. Here, by tracking the replisome β-clamp subunit in live *Caulobacter crescentus*, we find that rapid DNA segregation can give rise to a second focus which resembles a replisome, but does not replicate DNA. Sister replisomes can remain colocalized, or split apart to travel along DNA separately upon disruption of chromosome inter-arm alignment. Furthermore, chromosome arm-specific replication-transcription conflicts differentially modify replication speed on the two arms, facilitate the decoupling of the two replisomes. With these observations, we conclude that the dynamic chromosome organization flexibly shapes the organization of sister replisomes, and we outline principles which can help to reconcile previously conflicting models of replisome architecture.

Bacterial chromosomes are organized in a hierarchical manner by supercoiling, cohesion, and macrodomain formation[1]. This organization condenses the chromosome to fit into a cell orders of magnitude smaller than its own decompacted dimensions, with its genomic loci arranged into predictable cellular positions according to their location along the chromosome[2–4]. Chromosome replication and segregation take place concurrently with other processes such as transcription and repair, all acting on the same DNA template. Chromosome structure is both a consequence and a determinant of how these distinct functions are carried out[5], for instance: boundaries of chromosome interaction domains are enriched at highly expressed genes[6]; replication-transcription conflicts lead to replisome disassembly[7]; and DNA damage slows down replisome activity[8].

For most bacteria that possess a single circular chromosome, DNA replication initiates at the chromosomal origin of replication (*ori*) where two replisomes—multiprotein machines—are loaded.

Each replisome duplicates one arm, and they progress bidirectionally until meeting at the chromosomal terminus site (*ter*) and finally disassembling once replication is complete. Several competing models have been proposed for the organization of the two replisomes during replication progression, and developed over the past decades as evidence has accumulated. In the factory model, the two replisomes are physically associated, and may either be anchored in one location as a stationary factory, or move together as a mobile factory. A replication factory model was proposed in many bacterial species (*Escherichia coli*, *Bacillus subtilis*, *Pseudomonas aeruginosa*, *Helicobacter pylori*, and *Mycobacterium smegmatis*)[9–14], with the evidence that replisomes reside preferentially at certain locations within the cell, or appear as a single focus in fluorescence imaging. In the alternative track model, the two replisomes move independently along the two chromosomal arms. In support of this model, studies in *E. coli*, *B. subtilis*, and *Myxococcus xanthus* report splitting of sister replisomes into two

[1]Laboratory of Experimental Biophysics, Institute of Physics, Ecole Polytechnique Fédérale de Lausanne (EPFL), Lausanne, Switzerland. [2]National Centre for Biological Sciences, Tata Institute of Fundamental Research, Bangalore, India. [3]Department of Fundamental Microbiology, Faculty of Biology and Medicine, University of Lausanne, Lausanne, Switzerland. ✉e-mail: suliana.manley@epfl.ch

distinct foci[15–18]. Some of these studies may be partially confounded by multi-fork replication, in which more than one pair of replisomes simultaneously duplicates the genome to enable a competitive advantage by allowing cell cycle times to be shorter than chromosome replication times. Yet, contradictory results are also reported in *C. crescentus*, which strictly undergoes a single round of replication per cell cycle. A single focus of the replisome subunit clamp loader (HolB) was detected throughout the cell cycle[19], whereas two foci of the β-clamp (DnaN) were occasionally resolved in a subpopulation of cells[20].

A parsimonious explanation would be that either a factory or a track model is possible, but which is observed may depend on the context. This implies that, if the principles are well-understood, it should be possible to convert from one organization to the other. Here, we address the long-standing controversy of replisome organization by measuring the dynamics of replisomes and the duplication of genomic loci by time-lapse imaging, leveraging *C. crescentus* as a model system. We find that variations in chromosome segregation speed can lead to patterns in replisome localization which appear as two separate foci, but of which only one contains both functional replisomes. Furthermore, we demonstrate through genetic perturbations and genome rearrangements that chromosome inter-arm alignment acts as an indirect linker to maintain replisome colocalization. Finally, we observe that the positions of highly-expressed genes can affect replisome progression speeds in an arm-dependent manner and therefore lead to replisome decoupling - and that modifying this can alter replisome splitting dynamics. Collectively, our results demonstrate how chromosome organization impacts the cohesion and dynamic progression of sister replisomes, allowing for inter-conversion between co-localized versus independent replisome organization.

## Results

### Two patterns of DnaN dynamics during *C. crescentus* chromosome replication

To follow the dynamics of the replisomes, we imaged the replisome β-clamp subunit (DnaN) fused to superfolder GFP (sfGFP)[21] with time-lapse microscopy. The expression of DnaN-sfGFP from its native locus provided negligible photobleaching when imaged every 2 min, and a sufficient signal-to-noise ratio (SNR) to visualize its distribution for the cell cycle duration. In individual cells, DnaN went from diffuse to a single bright focus, indicating the onset of replisome assembly. We sometimes observed a dim DnaN streak or focus extended away from the bright one towards the opposite pole (Fig. 1a and Supplementary Movie 1), which appeared soon after replication initiation (Fig. 1b). In some cells, a single bright DnaN focus split into two foci near the end of replication, before merging and disassembling into a diffuse cytoplasmic signal (Fig. 1a, b). Thus, we refer to early splitting and late splitting as two distinct events in replisome organization deviating from a single focus. Overall, we observed early splitting in a large fraction (~85%) of cells (Supplementary Fig. 1), where the duration and relative position of dim DnaN signals varied considerably. Interestingly, we also found late splitting in ~38% of cells (Supplementary Fig. 1), which lasted for 11.9 min (10.7% of the mean replication duration) on average. We were able to observe these brief but striking events because the DnaN-sfGFP used here offers good photostability over high repetition and long duration time-lapse imaging.

To visualize DnaN organization at the population level, we plotted demographs composed of a kymograph of each cell, ordered by cell length at replication initiation (Fig. 1c). Since replication duration varies from cell to cell according to a normal distribution, we normalized time for each cell by its replication duration to display equivalent points in the replication cycle: 0% corresponds to the frame before a replisome focus appears, and 100% corresponds to the frame after it disassembles. We also normalized each kymograph to take on the full scale of an 8-bit image (between 0 and 255). In nearly all cells,

we observed a single DnaN focus proximal to the old, stalled pole, where the *ori* locus is located during the G1-to-S phase transition in *C. crescentus*[22], indicating replication initiation (1% replication time). A few cells (5.3%) exhibited initiating replisomes in other cellular locations (Supplementary Fig. 2), indicating the mispositioning of chromosomal *ori*[23]. Subsequently, in some cells a second dim focus or streak appeared at the early- to mid-replication stages (10–50% replication) and two foci emerged at the late stage (90% replication). We then plotted the raw intensity profiles of DnaN fluorescence at 30% and 90% replication time, to visualize the early splitting and late splitting events. We found that during early splitting events, the bright and dim foci were located further apart relative to late splitting events. Late-splitting foci had more similar intensities, although intensity varied considerably at both timepoints (Fig. 1d). In both cases, the positions of the two foci seemed to be symmetric relative to mid-cell in the majority of cells (Supplementary Figs. 3 and 4).

From the kymographs and intensity profiles, during replication DnaN typically appeared qualitatively either as a single bright focus, a bright and a dim focus or streak, or two bright foci of similar intensity. To characterize these patterns as a function of replication stage, we detected DnaN foci and measured their integrated intensities. Cells with two DnaN foci represented 10–30% of the total population at each timepoint, with peaks in the number of cells having two foci occurring early and late in replication (Fig. 1e and Supplementary Fig. 5). In cells with two foci, we also measured the intensity ratio, which exhibited a large variation during early- to mid-replication (20–60% replication time), but less variation during late replication (around 80–90% replication time). Using 1.5 as the threshold intensity ratio to classify the similarity of the two DnaN foci, asymmetric foci mostly appear at early replication stages and symmetric foci at later stages (Fig. 1f), consistent with the qualitative observations from kymographs.

### Late-splitting events revealed by other replisome components

For further insights into overall replisome dynamics beyond DnaN, we turned to additional replisome components. During DNA replication, single-stranded DNA binding (SSB) proteins prevent secondary structure formation. We performed time-lapse microscopy of SSB fused to sfGFP. In the early replication stage, SSB rarely formed a second focus as found for DnaN; and dim SSB signals appeared in kymographs as a diffuse background rather than a trajectory originating from the bright replisome focus (Fig. 2a i, Supplementary Fig. 6, and Movie 2). In contrast, late-splitting events of SSB were apparent in some cells (8.6 ± 3.1%) (Fig. 2a ii and Supplementary Fig. 6). To characterize SSB colocalization with DnaN, we imaged a dual-labeled strain with DnaN-sfGFP and SSB-mScarlet-I every 10 seconds for 10 frames or every 2 minutes for 3 h. We found that when dim DnaN foci were present, no equivalent signals were present in the SSB images (Fig. 2b i and Supplementary Fig. 7). However, among cells with two bright DnaN foci, most of them (88.1 ± 6.2%) also had two bright SSB foci, with which they colocalized (Fig. 2b ii and Supplementary Fig. 8, 9).

We also imaged the replisome subunits HolB (delta prime subunit of polymerase III) and DnaB (helicase). Both HolB and DnaB proteins are present at the replication fork in low abundance[24]. Time-lapse microscopy images of the YFP-fused HolB or DnaB exhibited a too low SNR to determine whether dim signals (i.e. early-splitting events) exist (Supplementary Fig. 10). However, we observed two bright HolB or DnaB foci from snapshots, indicating the appearance of late-splitting events (Supplementary Fig. 11). We further simultaneously imaged HolB-YFP with SSB-mScarlet-I (Fig. 2c), and found that for cells containing two bright HolB foci, two SSB foci also existed in a majority of them (90.6 ± 5.2%), with good colocalization (Supplementary Fig. 12). Similar results were obtained when we colocalized DnaB with SSB (Fig. 2d): for cells containing two bright DnaB foci, most of them (91.4 ± 4.3%) also showed two well-colocalized SSB foci (Supplementary Fig. 13). Collectively, we found the dim signal at early replication

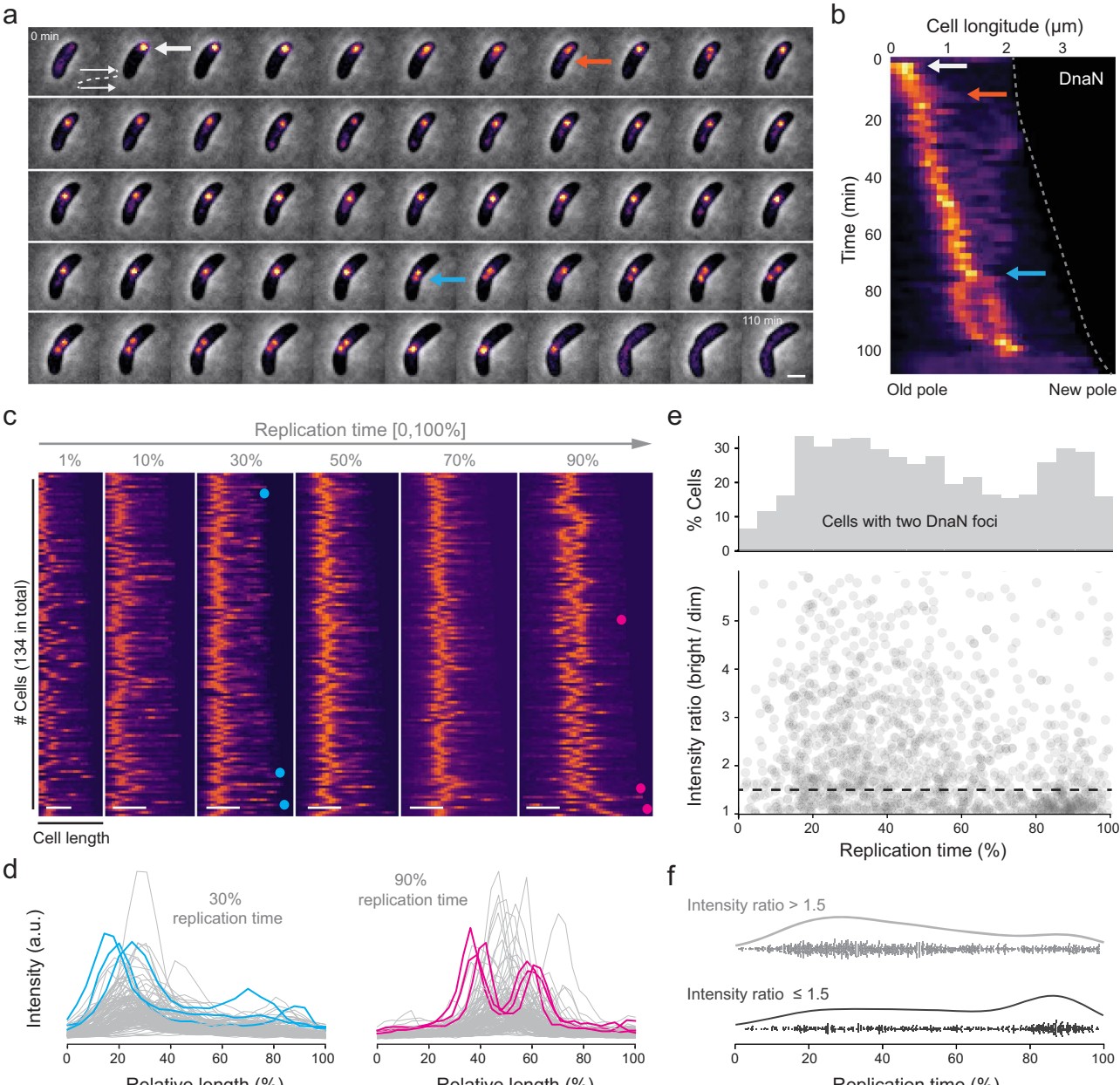

**Fig. 1 | DnaN dynamics over cell cycle. a** A montage of phase contrast and DnaN-sfGFP fluorescence (pseudo-colored as inferno) images for a single representative cell during replication with 2 min. intervals. **b** Kymograph of time-lapse imaging of the representative cell in **a**. Intensity profiles are aligned with old poles. Cell boundaries over time are highlighted by dash line. Replication initiation, appearance of early-splitting and late-splitting events are indicated by white, orange and blue arrows respectively in **a**, **b**. **c** Demograph of normalized DnaN-sfGFP fluorescence at different time points of normalized replication duration. **d** Intensity profiles of the demograph at 30% and 90% replication time in **c**. Three example profiles for each category are highlighted in blue or pink, with their positions indicated in **c**. **e** Frequency of cells that contain two detected DnaN foci (top; $n = 1641$), and the intensity ratio of bright/dim focus (bottom) as a function of replication time. Dashed line indicates the threshold value of 1.5. **f** Occurrence of cells with dissimilar ($>1.5$; $n = 1092$) or similar ($\leq 1.5$; $n = 549$) intensity of the two DnaN foci as a function of replication time. Source data underlying **d**–**f** are provided as a Source Data file. Scale bar in **a**, **c**: 1 μm.

stages (i.e. early-splitting events) to be specific to DnaN, while the two bright foci (i.e. late-splitting events) typically contained four different replisome components by pairwise co-localization.

## Early-splitting results from residual DnaN binding on rapidly segregated newly replicated DNA

The absence of additional replisome components called into question whether the dim DnaN structure observed following early-splitting events represented a functional replisome. Previously, similar data was interpreted as evidence for the "track" model of replisome progression[20]. To investigate the functionality of this DnaN-enriched

site, we tested whether it was the location of genomic locus duplication. We labeled two *ori*-proximal genomic loci (L1 & R1) on left and right arms respectively using orthogonal ParB/*parS* systems[25] (Fig. 3a). Since bright and dim DnaN foci were located at opposite cell halves at the early-replication stage, we expected to observe L1 and R1 moving to different cell halves prior to their duplications if each focus corresponded to a functional replisome (Fig. 3b). However, time-lapse montages and kymographs generally showed the trajectories of left- and right-arm *ori*-proximal loci residing at the same cell half until loci-splitting events occurred (Fig. 3c and Supplementary Fig. 14), indicating replication at colocalized replisomes, followed by segregation

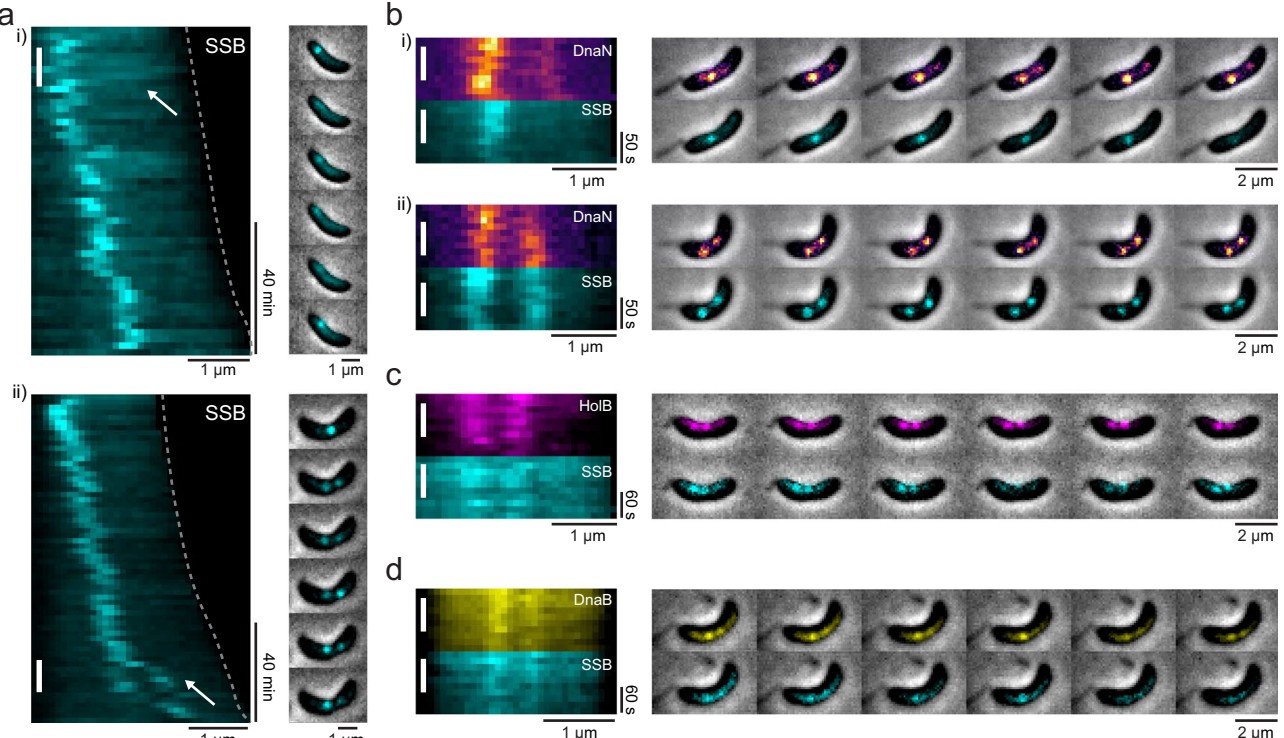

**Fig. 2 | Visualization of different replisome components. a** Kymographs and montages of two representative cells (i-ii) expressing SSB-sfGFP (pseudo-colored as cyan) with 2 min. intervals. Cell boundaries over time are highlighted by dashed lines. Appearance of diffuse SSB background (in i) and two bright SSB foci (in ii) are highlighted by arrows. **b–d** Kymographs and montages of representative cells expressing both DnaN-sfGFP and SSB-mScarlet-I (cyan) **b**, or both HolB-YFP (magenta) and SSB-mScarlet-I **c**, or both DnaB-YFP (yellow) and SSB-mScarlet-I **d**. The time interval is 10 s. for **b** and 15 s. for **c**, **d**. Vertical lines in kymographs **a–d** correspond to the time ranges over which individual images are shown as montages.

of one copy from this common location towards the opposite, new cell pole. In all, $84.7 \pm 0.5\%$ of cells exhibited dim DnaN signals at early replication stages, while $96.2 \pm 4.2\%$ of cells showed colocalized L1/R1 splitting sites, suggesting that the dim DnaN structures are not functional replisomes.

Dim DnaN signals usually appeared early in replication, so we hypothesized that their formation could be related to DNA segregation, which is initiated by the ParB-mediated partitioning of the *ori*-proximal *parS* sites from the old to the new cell pole[26,27] (Fig. 3d). We performed two-color imaging of DnaN and ParB in wild-type (WT, CB15N) cells, and found that dim DnaN foci appeared after the initiation of ParB migration (Fig. 3e). We also observed that the timing of ParB segregation matched the appearance of dim DnaN signals in kymographs (Fig. 3f i-ii and Supplementary Fig. 15), indicating a link between the two processes. As previously indicated, around 15% of cells maintained a single DnaN focus, which was unperturbed by segregation (Fig. 3f iii). To further investigate the difference between cells with a single focus and those with a second dim focus or streak, we considered the segregation rate. We defined the ParB segregation time as the period beginning with the formation of a DnaN focus at the old pole, and ending with the full migration of a ParB focus to the new pole. We found that the ParB segregation time for cells with dim DnaN signals was on average $12.6 \pm 4.4$ min, as opposed to $29.8 \pm 8.1$ min for those with a single DnaN focus (Fig. 3g). Therefore, dim DnaN signals could correspond to residual replisome molecules bound to DNA behind the replication fork, which haven't yet fallen off during rapid segregation. They may appear as a streak when the DNA they are attached to is extended, or as a punctum as the DNA compacts. The lack of dim signals may occur when the segregation time is much longer than the DnaN dissociation time, allowing time for full unloading from the segregating DNA. Notably, the dissociation time of

per DnaN clamp ($\beta$2 dimer) measured in living *E. coli* cells can be up to minutes long (0.78 or 2.75 min on average[28,29]), where more than half of the molecules in the cell were bound to DNA. We did not find a difference in total replication time between cells with versus without dim DnaN signals (Supplementary Fig. 16), suggesting that the speed of DNA segregation does not affect the overall replication progression.

If DNA segregation was driving the appearance of the dim DnaN focus or streak, disrupting segregation should diminish its formation. We thus imaged DnaN and ParB in *parAK20R* mutant cells, which are only capable of partial DNA segregation[30] (Fig. 3h). In this mutant background, we did not observe dim DnaN signals in single-cell montages or kymographs (Fig. 3i, j and Supplementary Fig. 17). Comparing demographs at early stages (30% replication) for WT and *parAK20R* confirmed that in segregation-deficient cells, DnaN appeared as a single focus, albeit a less compact one than in WT (Fig. 3k). We also detected DnaN foci in WT and *parAK20R* background cells, and found that only 0.8% of *parAK20R* cells contained two foci (with symmetric or asymmetric intensities), which is reduced ~10 times compared to the WT (Supplementary Fig. 18). Collectively, these results indicate that the dim DnaN focus or streak is not an active replisome, but likely corresponds to residual binding on newly segregated DNA behind a replication fork. Thus, the single bright focus at early- and mid-replication stages represents two colocalized, active replisomes.

## Chromosome inter-arm alignment maintains the colocalization of sister replisomes

A single focus contained both replisomes for most of the replication cycle; however, late-splitting events often produced two foci of similar intensity. In the absence of a known protein linker, we hypothesized that chromosome inter-arm alignment may offer an indirect mechanism to maintain the colocalization of the two replisomes (Fig. 4a).

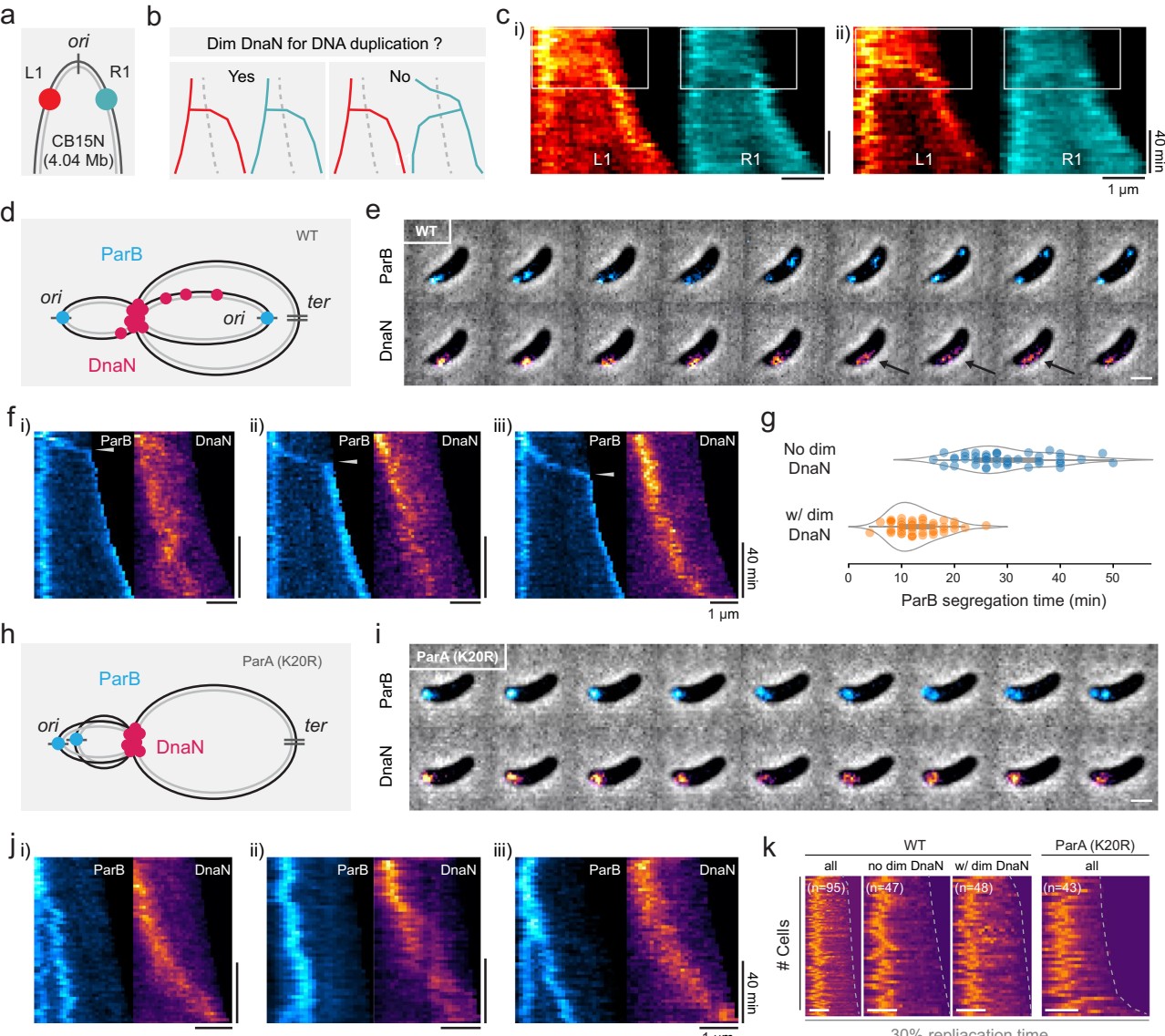

**Fig. 3 | Impact of DNA segregation on DnaN localization. a** Schematic of labeled *ori*-proximal loci (L1 and R1) on left and right arms. **b** Expected L1 and R1 kymographs should have different trajectories depending on whether the dim DnaN signal is a functional replisome involved in DNA duplication. **c** Two representative kymographs of L1/R1 fluorescence with 4 min. intervals. Identical scale bars for i-ii. **d** Schematic of DNA-bound ParB and DnaN molecular distributions in WT cells. **e** A representative montage of ParB and DnaN fluorescence in WT cells taken at 2 min. intervals. Dimmer DnaN streaks are highlighted in arrows. **f** Kymograph examples of DnaN and ParB in WT during early replication, identical scale bars for i-iii. Time points of ParB segregation completion are highlighted in arrows. **g** ParB segregation time for cells with (*n* = 48) or without (*n* = 47) a dim DnaN structure. Source data are provided as a Source Data file. **h** Schematic of DNA-bound ParB and DnaN molecules in segregation-deficient *parAK20R* cells. **i** A representative montage of ParB and DnaN fluorescence in *parAK20R* taken at 2 min. intervals. **j** Kymograph examples of DnaN and ParB in *parAK20R* during early replication, identical scale bars for i-iii. **k** Demographs of DnaN fluorescence at 30% replication time for (left to right): WT (all cells, w/o dim DnaN, w/ dim DnaN) and all *parAK20R* cells. Scale bar in **e, i, k:** 1 μm.

In *C. crescentus*, inter-arm alignment is enabled by Structural Maintenance of Chromosome proteins SMC, ScpA, and ScpB, which load at the *ori*-proximal *parS* and cohere ~600-kbp proximal regions of the left and right chromosomal arms[31–33]. In this model, splitting of the two replisomes could result from diminished alignment.

To test this idea, we used an *smc* knockout strain (*Δsmc*), which was reported to have reduced inter-arm alignment compared to WT cells[33] (Fig. 4b). SMC is a non-essential protein in *C. crescentus*, and the *Δsmc* strain shows normal replisome progression and division, but slightly slower cell growth[31,33]. Kymographs of DnaN fluorescence revealed that many *Δsmc* cells (41.8 ± 1.9%) contained two bright foci at early- to mid-replication time (Fig. 4b, Supplementary Fig. 19 and Movie 3), which was rarely observed in WT cells (6.9 ± 3.9%) (Supplementary Fig. 1).

We also studied a strain with partial chromosome inversion, between +3611 and +4038 kb (*flip1-5*)[33], which contains an ectopic *parS* site on the left arm (~427 kb away from *ori*) and leads to altered inter-arm alignment flanking the site (Fig. 4c). In contrast to WT and *Δsmc*, splitting occurred in most *flip1-5* cells (68.5 ± 8.2%) (Fig. 4c, Supplementary Fig. 20, and Movie 4), into two DnaN foci with generally similar (within 2-fold), but fluctuating intensities (Fig. 4d). We also imaged SSB-sfGFP in *flip1-5* background cells every 2 min, and found such splitting appeared in most cells (94.9 ± 4.5%) at early-mid replication stages (Supplementary Fig. 21 and Movie 5). Overall, the *smc* knockout results in an intermediate phenotype between WT and *flip1-5* for foci-splitting events. In many cases, the two foci migrated to different cell halves then finally merged at the division site, resembling the behavior of late-splitting events in WT cells. Collectively, these results suggested that

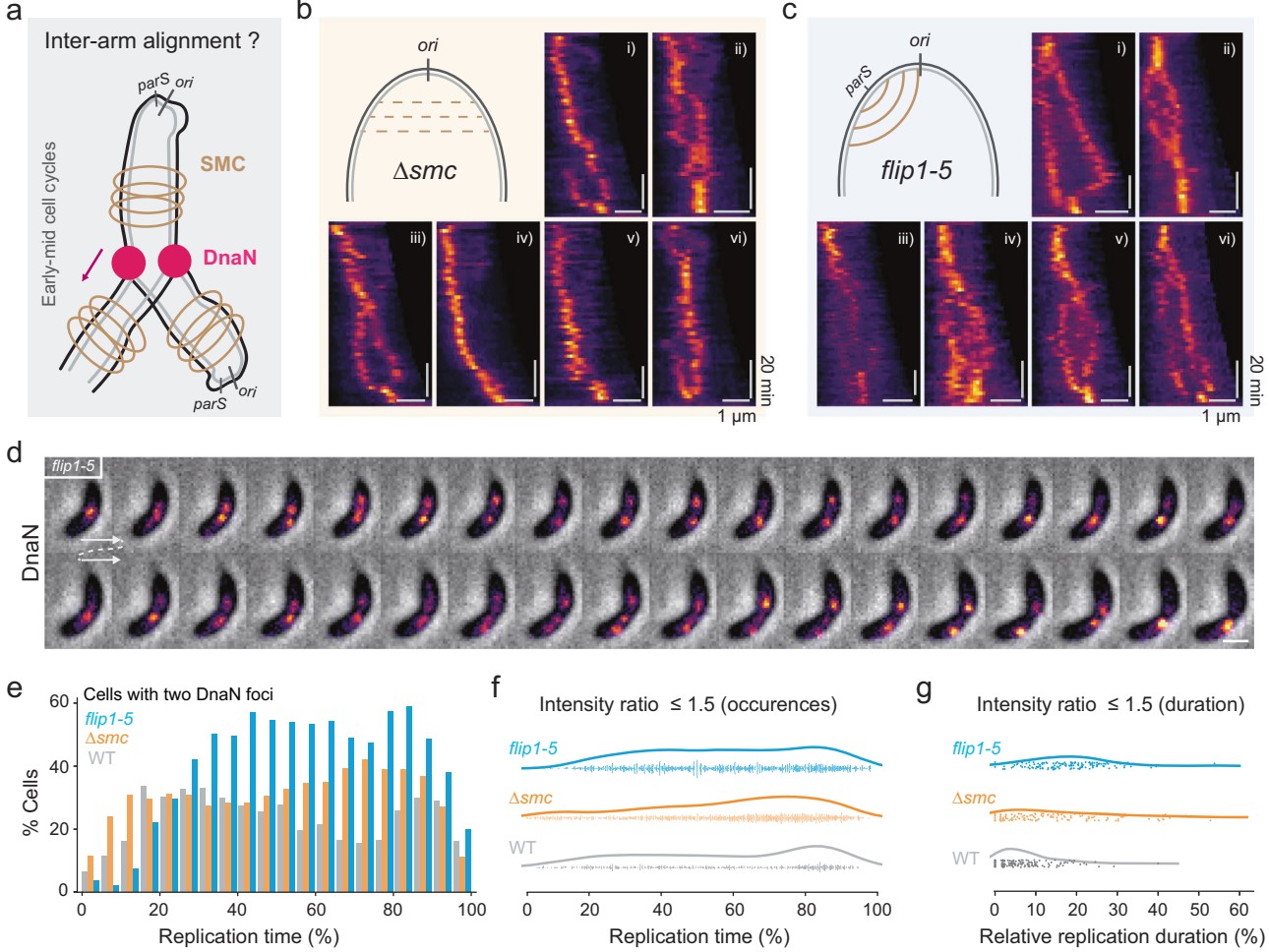

**Fig. 4 | Inter-arm alignment maintains colocalization of sister cohesion.**
**a** Schematic of inter-arm alignment by SMC on colocalization of two DnaN foci.
**b** Schematic of *Δsmc* cells with reduced inter-arm alignment, and corresponding DnaN kymographs. **c** Schematic of *flip1-5* cells with distorted alignment, and corresponding DnaN kymographs. **d** A representative montage of phase contrast and DnaN fluorescence taken at 2 min. intervals in *flip1-5* cells. **e** Distribution of cells that contain two detected DnaN foci in WT (*n* = 1641), *Δsmc* (*n* = 2575), and *flip1-5* (*n* = 2737) cells. **f** Occurrence of cells with similar DnaN foci (intensity ratio ≤1.5) as a function of replication time (*n* = 549; 1191; 1195 for WT; *Δsmc*; and *flip1-5* cells respectively). **g** Duration of DnaN foci with similar intensity relative to the overall replication time (*n* = 134; 133; 138 for WT; *Δsmc*; and *flip1-5* cells respectively). Source data underlying **e**–**g** are provided as a Source Data file.

disrupted inter-arm alignment at *ori*-proximal regions could decouple sister replisomes, leading to independent replisome progression in *C. crescentus* and switching from a "factory" to a "track" model.

We quantitatively analyzed the split DnaN foci in these different strains. Compared to WT, more *Δsmc* cells showed two DnaN foci at later stages (50–90% replication), while an even larger portion (~60%) of *flip1-5* cells exhibited two foci (Fig. 3e and Supplementary Fig. 22). Similar DnaN foci (intensity ratios ≤1.5) in *flip1-5* cells appeared with similar prevalence throughout most of the replication time, while in *Δsmc* and WT most appeared at late stages (~80% replication) (Fig. 3f). We also measured the lifetime of similar intensity DnaN foci relative to the total replication time. As expected, replisomes remained split longer on average in *flip1-5* (~20% replication) compared to WT (5% replication) and *Δsmc* (~8% replication) cells (Fig. 3g). Altogether, these results demonstrate that sister replisomes split earlier and more frequently in the absence of SMC protein, and that the disruption of inter-arm alignment can decouple the two replisomes from nearly the onset of replication.

### Late-splitting is coincident with progression mismatch between sister replisomes

We next dissected the phenomenon of late-splitting replisomes in WT cells. We imaged DnaN-sfGFP at an increased frame rate

(10 s/frame), and observed that one replisome often migrated to the future division site ahead of the other one (Fig. 5a and Supplementary Fig. 23). To provide a genomic context, we simultaneously imaged the chromosome terminus binding protein ZapT[34], and found that during late-splitting: (1) one DnaN focus advanced to the constriction site until pausing once colocalized with ZapT; (2) the second DnaN focus continued until arriving at the terminus site; (3) the two foci merged, before disassembling (Fig. 5b i-iv). We further tracked the two DnaN foci, and found that foci at division sites had a smaller mean step size (~25 nm), indicating relative confinement (Fig. 5c and Supplementary Movie 6). Colocalization of the replisomes with ZapT and reduced mobility prior to their disassembly could be expected, since ZapT is known to link the chromosomal terminus to FtsZ[34,35], which forms a ring-like scaffold for other proteins involved in cell constriction and division[36,37]. However, the asynchronous arrival of two replisomes at the *ter* site was intriguing.

We wondered whether one replisome preceded the other to the terminus due to inter-arm differences in DNA duplication rate. To test this idea, we labeled *ter*-proximal loci on both left and right arms using orthogonal ParB/*parS* systems, but were unable to visualize locus duplication due to insufficient SNR (Supplementary Fig. 24). As an alternative, we used strains with individual genome loci labeled via the fluorescent reporter operator system (FROS)[2]. Our ten strains

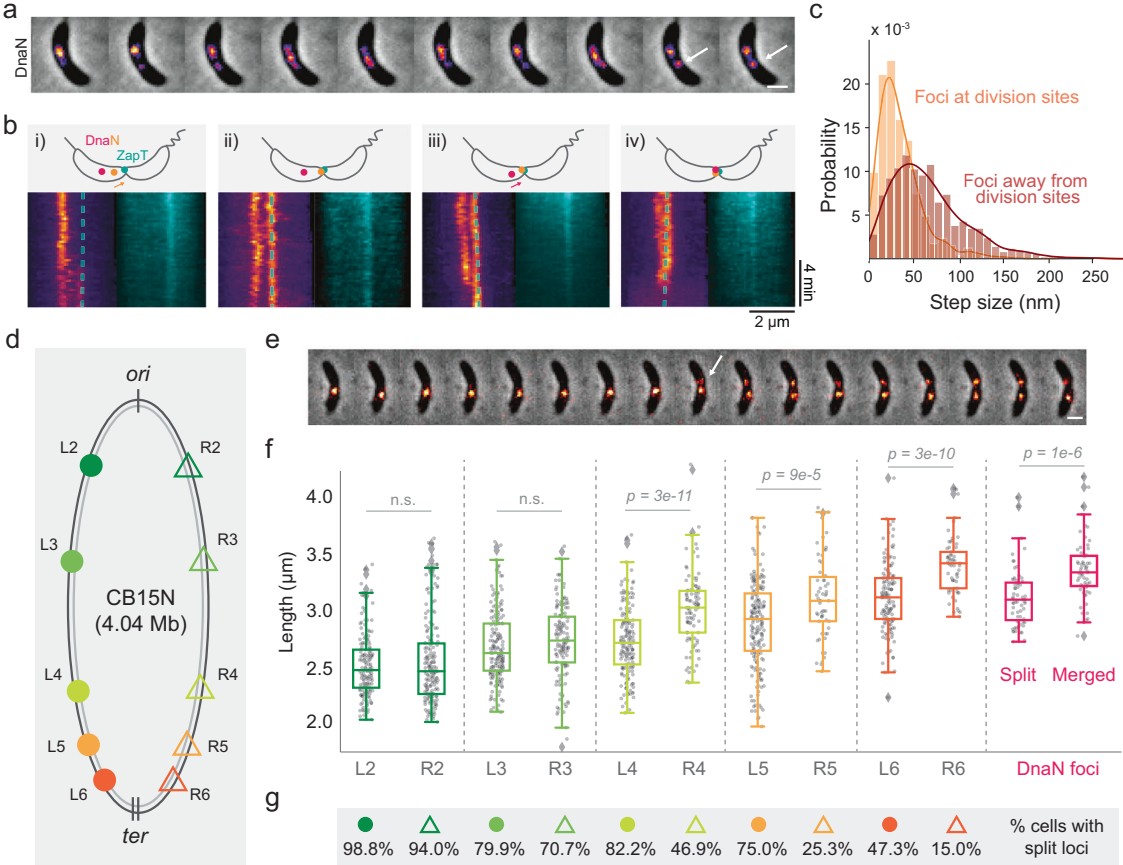

**Fig. 5 | Replisomes arrive at *ter* asynchronously. a** A representative montage of DnaN fluorescence in pre-divisional cells (arrows highlight constriction sites) taken at 10 s. intervals. **b** Schematic of replisome (DnaN) and ZapT positioning (upper), and corresponding kymographs in four example cells (**i-iv**, dashed lines in DnaN kymographs indicate ZapT location). **c** Step size probability distributions for late-splitting DnaN foci, classified by their position either at ($n = 1112$) or away from the division site ($n = 1107$), normalized to a total probability of 1. **d** Schematic of ten genomic loci labeled in individual strains (L2/R2: −0.37/ +0.43 Mb; L3/R3: −0.99/ +0.98 Mb; L4/R4: −1.38/ +1.38 Mb; L4/R5: −1.52/ +1.52 Mb; L6/R6: −1.67/ +1.69 Mb). **e** A representative montage of the L6 locus replication (segregation highlighted by arrow) taken at 2 min. intervals. **f** Length distribution of cells at the moment of genomic loci segregation and DnaN foci-splitting/merging ($n = 168$; 186; 187; 143; 176; 98; 192; 58; 149; 61; 63 for L2; R2; L3; R3; L4; R4; L5; R5; L6; R6; and DnaN foci/ loci-splitting events, respectively). Data are presented by boxplot where the minima, maxima, center, and bounds of box indicates the lowest value, highest value, median value ($50^{th}$ percentile), and interquartile range (IQR, $25^{th}$ to $75^{th}$ percentile) of the data, respectively. Lower or upper whisker extend from the edge of box is defined by 1.5 times of the IQR range, and data points beyond this range are considered as outliners. **g** The percentage of loci-splitting events for each strain. Scale bar in **a**, **e**: 1 μm. The *p*-value is calculated by a two-tailed *t*-test. Source data underlying **c**, **f** are provided as a Source Data file.

correspond to five pairs of loci labeled on the left or right arm (e.g. L2 or R2) with a similar distance to *ori* (Fig. 5d). The same FROS (*tetO*/ TetR-YFP) was used in all strains to achieve similar SNR, and all strains had consistent cell lengths post-synchrony[2]. In time-lapse imaging, we identified the frame where each genome locus split—which indicated locus duplication. Taking cell length as a proxy for cell age (Fig. 5e), we could then determine when each genome locus was duplicated.

We compared the length— a proxy for time—at locus duplication for the ten strains (Fig. 5f). Both L2/R2 and L3/R3 pairs were duplicated at similar times on average, suggesting that sister replisomes progressed at same rates within these chromosomal regions. However, a significant difference emerged in the duplication of the L4/R4 pairs, with R4 on the right arm duplicated later than L4 on the left arm. We also observed lags for L5/R5 and L6/R6 pairs in an arm-specific manner: DNA replication on the right arm was significantly delayed at *ter*-proximal regions compared to loci on the left arm at similar genomic positions relative to *ori* and *ter*. Interestingly, we found locus splitting was undetectable in some cells, although they divided normally; also, split loci sometimes existed for only few minutes and then disappeared (Supplementary Fig. 25). We quantified that in each pair, loci labeled on right arms were less likely to exhibit splitting compared to those on left arms, especially at *ter*-proximal regions (Fig. 5g). We suspect that the

absence of loci splitting results from the removal of TetR-YFP on *tetO* arrays, potentially driven by competition with other DNA binding proteins such as those involved in DNA transcription or packaging. We also noticed that, on average, replisome late splitting occurred after the duplication of R3 locus, and merging when the R6 locus was almost duplicated (Fig. 5f). Overall, the late splitting of sister replisome was coincident with asynchronous DNA duplication between the two chromosome arms, suggesting that the decoupling of sister replisomes could be related to replication barriers on the right arm.

## Highly transcribed genes promote replisome decoupling
We suspected that arm-specific replication delays resulted from replication-transcription conflicts, during which both DNA replication and transcription (TX) bind to the same template concurrently. We surveyed highly transcribed genes located between R3 (+ 0.98 Mb) and R6 (+ 1.69 Mb) regions, where replisome splitting appeared. Two candidate genes reside there: *rsaA* (+1.16 Mb) and the *rDNA* gene cluster (+ 1.43 Mb). The *rsaA* gene encodes the S-layer protein in *C. crescentus* which accounts for ~31% of its total proteome[38], while *rDNA* encodes the essential ribosomal translation machinery for protein synthesis (Fig. 6a). Furthermore, quantitative analysis of next-generation sequencing data revealed that replication progression could pause at

the *rDNA* site in *B. subtilis*[39]. Therefore, we hypothesized that replication-TX conflicts on the right arm played a role in splitting sister replisomes.

We investigated replisome dynamics in strains with manipulations of the *rsaA* gene, but did not perform any modifications to *rDNA* due to its essentiality. A previous study reported that the translocation of *rsaA* did not affect cell viability but reshaped chromosome domain formation at the ectopic site[6]. Yet, kymographs of DnaN fluorescence in *rsaA* knockout (*ΔrsaA*) cells exhibited high cell-to-cell variability (Fig. 6b, Supplementary Fig. 26 and Movie 7), without any obvious patterns in the replisome dynamics. Similar results were found in an *rsaA* translocated strain (*ΔrsaA::P_{xyl}-rsaA*), where the native *rsaA* gene was moved to the xylose-inducible promoter (*P_{xyl}*, +0.95 Mbp) site on the right arm (Supplementary Fig. 27). We suspected that the deletion of the *rsaA* gene at its native site led to instability in chromosome organization. Thus, we constructed a *rsaA* merodiploid (*rsaA+*) strain by inserting a second copy of *rsaA* at the *P_{xyl}* site, in addition to the native copy. Replisomes in *rsaA+* cells exhibited more regular patterns (Fig. 6c, Supplementary Fig. 28 and Movie 8), and many exhibited splitting of two bright DnaN foci at mid- and late-replication stages (~70%). The translocated *rsaA* was inserted with its transcriptional orientation from *ter* to *ori*, opposite to the native gene, introducing in head-on replication-TX conflicts. In some *rsaA+* cells, replication termination did not colocalize with the division site (Fig. 6d and Supplementary Fig. 29), potentially because

the additional *rsaA* gene affected chromosome organization and the positioning of the *ter* site.

We then quantified the foci splitting events in *rsaA+* strains. Compared to the WT, dramatically more *rsaA+* cells contained two DnaN foci throughout replication (Fig. 6e and Supplementary Fig. 30), although equally bright foci (intensity ratio ≤1.5) mostly occurred at the late stage (60–100% replication) (Fig. 6f). We further measured the first occurrence of late-splitting events in WT and *rsaA+* manually to compare the onset of DnaN splitting, and found that in *rsaA+* cells it occurred earlier than in most WT cells (Fig. 6g). Therefore, by inserting an additional *rsaA* gene ~23.8% closer to the *ori* (from +1.43 to +0.95 Mbp) on the right arm (*ter*: +2.02 Mbp), the onset of the late splitting DnaN foci appeared ~13.6% earlier on average relative to the replication time. Collectively, these results demonstrate that arm-specific replication-TX conflicts can influence the decoupling of sister replisomes.

## A model of replisome dynamics regulated by chromosome organization

Our study reveals that chromosome segregation can impact replisome protein organization, while compaction and replication-TX conflicts play roles in coupling and decoupling replisomes. Based on these data, we propose a model to describe replisome dynamics in *C. crescentus* cells (Fig. 7). Initially, sister replisomes assemble at the *ori* and proceed bidirectionally while duplicating the DNA—the two replisomes colocalize, and are indirectly tethered via inter-arm alignment enabled by

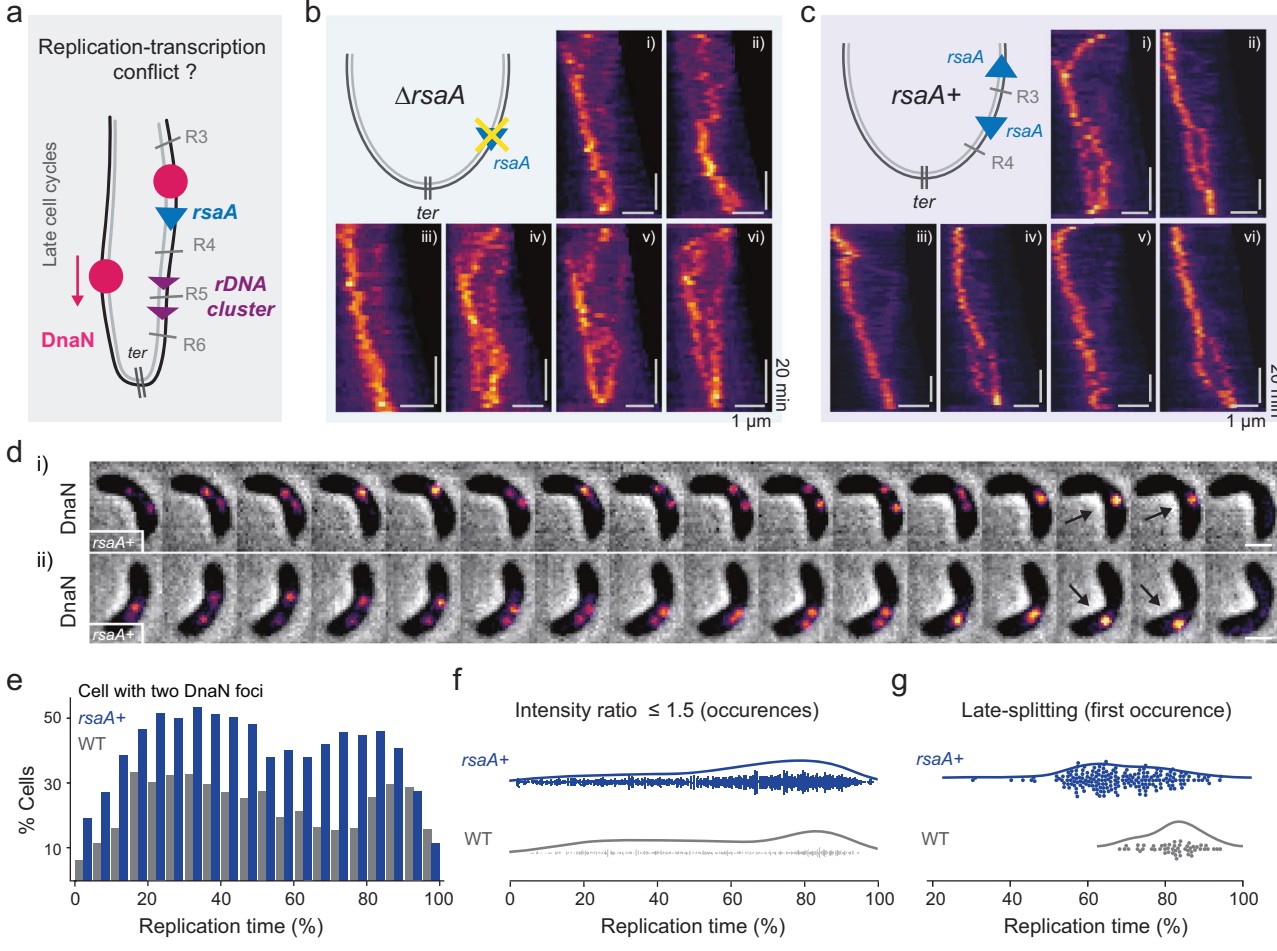

**Fig. 6 | The *rsaA* gene position affects replisome splitting. a** Schematic of potential replication-transcription conflicts. **b** Replisome dynamics in *rsaA* knockout cells (*ΔrsaA*). **c** Replisome dynamics in cells harboring an additional *rsaA* copy (*rsaA+*). **d** A representative montage of time lapse phase contrast and DnaN fluorescence taken at 2 min intervals in *rsaA+* cells. Scale bar: 1 μm. **e** Frequency of cells that contain two detected DnaN foci in WT (*n* = 1641) and *rsaA+* (*n* = 6948) cells. **f** Occurrence of cells with similar DnaN foci (intensity ratio ≤ 1.5) as a function of replication time (*n* = 549; 2114 for WT and *rsaA+* cells respectively). **g** Time of first occurrence of late-splitting into two replisomes (*n* = 51; 201 for WT and *rsaA+* cells respectively). Source data underlying **e**–**g** are provided as a Source Data file.

Structural Maintenance of Chromosome proteins (Fig. 7a). Replication on both arms proceeds at similar rates, and newly replicated DNA is segregated to the poles and compacted. If DNA segregation to the new pole occurs rapidly, some of the DnaN molecules which remain bound behind the replication fork travel with the nascent DNA strand, forming a streak-like signal (Fig. 7b). DnaN is a sliding clamp, so it can slide along the DNA as it is compacted, and accumulate into a weaker focus resembling a second replisome near the new pole; however, it is not capable of replicating DNA. Meanwhile, colocalized sister replisomes are indistinguishable under diffraction-limit microscopy.

Since sister replisomes are only indirectly tethered, they are susceptible to decoupling into separate foci if one progresses substantially slower than the other. This can occur when one arm contains more highly transcribed regions, since the replisome that encounters stronger replication-TX conflicts may slow its progression (Fig. 7c). Such a mismatch in replication rates would further disrupt local inter-arm alignment and generate a less compacted chromosome region in the vicinity of the replisomes, allowing sister replisomes to be split apart by entropic forces. The faster-progressing replisome can then continue to the terminus in a solitary manner, where it pauses until being rejoined by the slower replisome (Fig. 7d), after which both disassemble.

## Discussion

Our study underlines that understanding replisome organization requires taking into account chromosome organization and template accessibility. In *C. crescentus*, asymmetric chromosome segregation leads to replisome protein patterns: a second dim focus or streak, previously interpreted as evidence for the track model. However, sister replisomes are initially colocalized and indirectly tethered via chromosome inter-arm alignment, and in most cases remain together until termination of replication, consistent with the factory model. It is still possible that unidentified proteins in *C. crescentus* could physically tether sister replisomes, but whose impact could be overcome by deformed chromosome structures (i.e. when SMC is knocked out or the *parS* site is translocated). Since chromosome segregation is highly

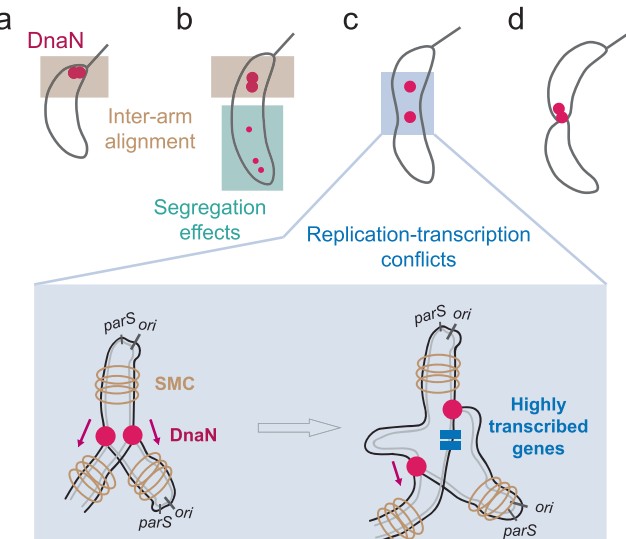

**Fig. 7 | Schematic illustrating the effects of chromosome organization on replisome dynamics. a** Sister replisomes assemble at the origin of replication together, and are indirectly coupled via inter-arm alignment. **b** At early- and mid-replication stages, remnant DnaN molecules residing on rapidly segregated DNA can resemble a second dim "replisome". **c** Replisomes can lag behind and eventually split after encountering arm-specific replication-transcription conflicts. **d** The faster replisome pauses at the terminus until joined by its sister, when both disassemble together.

organized across bacterial species, residual binding of replisome components behind the replication fork could be a confounding factor in interpreting the number of foci as a readout for replisome organization. This could be especially true in the case of the sliding beta-clamp (DnaN) component of the replisome, which can bind in excess and may slide on the newly-synthesized DNA following segregation and compaction, accumulating locally to form a second focus[40].

On the other hand, arm-specific replication-TX conflicts can cause replisome splitting, the dynamics of which is affected by the chromosomal position of highly transcribed genes. We estimated the time $\Delta t$ between replication of neighboring genomic loci based on a model of exponential length growth: $L''/L' = e^{\alpha \Delta t}$, where $L'$ and $L''$ represent the average cell length measured at splitting of the first and second locus. Based on that, we can calculate the replication speed ($\Delta bp/\Delta min$) between two loci given the genomic distance between them (Supplementary Fig. 31). We found that the right-arm replication speed slows down on average from R3 to R6 loci with some fluctuations, while the left-arm replication speed peaks at L3 and L4 followed by a decrease at L5 and L6 loci. This theoretical estimate suggests that when the right-arm replisome encounters replication-TX conflicts, it slows down, while replisome progression speeds up on the left arm. It is possible that replisome coupling tends to act as a resistance on the less conflicted arm, and once they are decoupled it can progress more rapidly. Given the conserved nature of chromosome organization, we expect that arm-specific conflicts between replication, transcription, and possibly repair should be ubiquitous across bacterial species. Indeed, a study in *E. coli* and *B. subtilis* revealed colocalization of sister replisomes in ~80% of cells, with the rest exhibiting two resolvable foci[17]. Although a "factory-like" model was proposed[41], a mixed model would make for a simple explanation, and would be consistent with our findings in *C. crescentus*. Thus far, several proteins including SeqA[42], CrfC[43], and MukB[44] have been proposed to facilitate colocalization of sister replication forks in *E. coli*; while the protein linker (Ctf4) in yeast that dimerizes the helicase of two replisomes provides the only direct molecular mechanism for assembling a replisome factory[45,46].

As a side note, we used two widespread tools—ParB/*parS* and FROS—to visualize genomic locus duplication. However, we found that the SNR of the orthologous ParB/*parS* system was dependent on the targeted chromosomal region, presumably due to differences in the abundance of ParB proteins bound the *parS* site. Lower abundance may be attributed to diminished accessibility. Similarly, the FROS system was affected by the target site, and we were surprised to observe an absence of genomic locus splitting in some cells (Fig. 5g). To date, both tagging methods have been widely applied in bacteria, yeast and mammalian cells[47], yet our results indicate that local genome architecture or competition for a common substrate may complicate data interpretation.

The advantages of a factory organization of the replisomes are largely presumptive—coupling the machinery together is hypothesized to lead to greater efficiency and higher fidelity of replication. A recent study in *E. coli* offered more direct evidence of potential benefits: disrupting colocalized replisomes with transcriptional roadblocks on one arm led to slowdown of the sister replisome, increased fork stalling and requirement of fork restart[9]. In *C. crescentus*, we measured the doubling time of WT, *Δsmc*, *flip1-5*, *ΔrsaA*, and *rsaA+* background strains in different nutrient conditions (Supplementary Table 1), and found that while they all grew similarly in minimal medium (M2G), *flip1-5* grew significantly slower in rich medium (PYE) compared to the others. Notably, the *flip1-5* strain also exhibited greatest disruption of replisome coupling (Fig. 4e-g). It is possible that in the *flip1-5* strain, replication-TX conflicts increase at *ori*-proximal regions, which contributes to the early splitting of DnaN foci and prolongs replication. In such a case, the doubling time of *C. crescentus* in rich medium could become limited by the chromosome replication time, because it is restricted to one round of replication per cell

cycle—and replisome coupling may confer an advantage by enabling more rapid population growth.

## Methods

### Bacterial strains and culture

All strains and plasmids used in this study are listed in Supplementary Data 1. Detailed protocols for the strain and plasmid construction are described in Supplementary Methods. A single colony of *C. crescentus* cells on the PYE (peptone, Merck, #82303; yeast extract, Merck, #Y1626) agar plate was inoculated into 3 ml of M2G medium to grow overnight at 28 °C under mechanical agitation (200 r.p.m.)[48]. Liquid cultures were re-inoculated into fresh M2G medium (OD$_{660}$ ~ 0.05) to grow cells until early exponential phase (OD$_{660}$ = 0.15–0.25) before imaging. Antibiotics (25/5 µg ml$^{-1}$ spectinomycin, 5/1 µg ml$^{-1}$ kanamycin) were added in solid/liquid cultures for selecting cells containing related antibiotic markers. To induce the expression of SSB-sfGFP/mScarlet-I, ZapT-mScarlet-I, and TetR-YFP under the $P_{xyl}$ promoter, 0.2% wt/vol xylose were added to the culture 2 h before imaging.

### Time-lapse microscopy

*C. crescentus* cell cultures were spotted onto a M2G agarose pad for time-lapse imaging. To make the agarose pad, a gasket (Invitrogen, Secure-Seal Spacer, S24736) was placed on a rectangular glass slide, and filled with 1.5% M2G agarose (Invitrogen, UltraPure Agarose, 16500100) solution without containing antibiotics. Another glass slide was placed on the top of the silicone gasket to make a sandwich-like pad. The pad was placed at 4 °C until the agarose solidified. After 20-40 min, the top cover slide was removed, and a 3-5 µl drop of cell cultures (OD$_{660}$ adjusted to ~0.2) was placed on the pad. After full absorption of the liquid, the pad was sealed with a plasma-cleaned #1.5 round coverslip of a diameter of 25 mm (Menzel). Imaging was performed at 28 °C on a customized Zeiss microscope (63× objective, NA 1.4) equipped with an autofocus system.

*C. crescentus* cells expressing DnaN-sfGFP were excited at 488 nm with 200 ms exposures using 6% of LED power (CoolLED, pE-800), together with phase contrast imaging with 200 ms exposures using 30% of transmission light. Cells expressing EGFP/YFP/mCherry(or mScarlet-I) fused proteins were excited at 488/510/560 nm with 200/200/200 ms exposures using 6/6/10% of LED power, respectively. Exclusively, cells expressing HolB-YFP or DnaB-YFP fusion proteins were excited at 510 nm with 1000 ms exposure using 50% of LED power. Custom emission filters (Chroma 89402 for DAPI/FITC/TRITC/Cy5 or 89403 for CFP/YFP/RFP/Cy7) were used when appropriate. Images were acquired by a CMOS camera (Teledyne Photometrics, Prime) with a 103 nm pixel size. Imaging data were collected through VisView 5.0.0.21.

### Imaging processing and analysis

Whole field-of-view image stacks were first corrected for drifting using the Fiji plugin MultiStackReg[49]. A transformation was obtained when registering phase contrast images, and then applied to the fluorescent channel. Single cell time-lapse image stacks during replication were cropped manually in time based on the appearance and disappearance of DnaN-sfGFP foci. Image stacks were used as inputs in MicrobeJ 5.13n(13) for both cell outline extraction and foci detection[50]. Intensity profiles for making kymographs were aligned with stalked poles, which is determined by the shape asymmetry of *C. crescentus* (i.e. in a pre-divisional cell, the future daughter cell connected with a stalk is longer than the opposite one which is connected to flagella). DnaN signals were detected as foci only if their shape was well-fit with a Gaussian. Information containing cell morphology (e.g. length, area) and foci properties (e.g. amplitude, sigmaX, sigmaY, offset) for each frame were generated and output in a results table. Parameters used for MicrobeJ processing were manually optimized and saved as a template

file, which is available in Zenodo together with the imaging data. The detected foci were further filtered by width (sigmaX or sigmaY values, between 1 and 3) and SNR (amplitude/offset ≥ 0.25). The integrated focus intensity was calculated as amplitude × sigmaX × sigmaY.

### Microplate reader experiments

Overnight cell culture of *C. crescentus* cells (OD$_{660}$ ~ 1) were dilute 1000 times into the fresh medium. A 200 µL of diluted culture was loaded into 96-well plate for measuring the growth curve. The microplate reader device (Thermo Scientific, Multiskan FC) was set on 28 °C or 32 °C under shaking conditions (200 rpm), and recorded the absorbance (A$_{660}$) every 20 min over 40 h. Five biological repeats were measured for each strain. To calculate the doubling time ($T_{db}$), the time ($\Delta T$) for cells growing from early-log phase (OD1, OD$_{660}$ ~ 0.1) to mid-log phase (OD2, OD$_{660}$ ~ 0.3) was count, and use the formula: $T_{db} = \ln2/(\ln(OD2/OD1))/\Delta T)$.

### Statistics and reproducibility

All samples were repeated at least in three biological duplicates. *C. crescentus* cells were grow in liquid M2G medium and then loaded onto the agarose pad for time-lapse imaging. Typically, thousands of cells in total were acquired for analysis. No statistical method was used to predetermine sample size. Detected DnaN foci failed with a Gaussian fitting, for instance, zero amplitude, extreme small (<1) or large (> 3) sigma values, or low signal-to-noise ratio (amplitude/offset <0.25), were excluded from the analyses. The experiments were not randomized, and the investigators were not blinded to allocation during experiments and outcome assessment.

### Reporting summary

Further information on research design is available in the Nature Portfolio Reporting Summary linked to this article.

## Data availability

The imaging data and spreadsheets containing manual records and annotations generated in this study have been deposited in the Zenodo database[51] [https://doi.org/10.5281/zenodo.10203421]. Source data underlying Figs. 1e, f, 3g, 4e–g, 5c, f, 6e–g, and Supplementary Figs. 16 and 31 are provided as a Source Data file. Source data are provided with this paper.

## Code availability

The running template for MicrobeJ 5.13n(13) and code for data post-processing (Python 3.7.3) used in this study have been deposited in the Zenodo database[52].

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

## Acknowledgements

We thank H. Perreten for technical support with cell culture and plasmid construction. We are grateful to P. Viollier, T. Le, X. Wang, M. Thanbichler and O. Shogo for gifts of *C. crescentus* strains and plasmids. This work was supported by the European Union's H2020 program under the European Research Council (ERC; CoG 819823 Piko, to S.M. and C.Z.) and Swiss National Science Foundation (grant 310030_204822 to J.C.). A.B. lab acknowledges support of the India Alliance Intermediate Grant (Grant number IA/I/21/1/505630).

## Author contributions

C.Z. and S.M. conceived the project. J.C., A.B and S.M. supervised the project. C.Z., A.M.J., L.C., J.C. A.B and S.M. designed experiments. C.Z. performed the imaging experiments and analysis; A.M.J. performed pilot experiments and preliminary data analysis; L.C. performed plasmid and strain constructions; and all coauthors discussed the results. C.Z. and S.M. wrote the manuscript with contributions from all authors. C.Z. prepared the figures.

## Competing interests

The authors declare no competing interests.
