## [Peer Review File · Nature Communications]

REVIEWER COMMENTS

Reviewer #1 (Remarks to the Author):

In most bacteria with the circular chromosome, DNA replication occurs bidirectionally from the unique origin, forming two sister replication forks. The replisome assembled on the fork represents a highly dynamic behavior, as demonstrated by the subcellular localization patterns of the replisome components, including the clamp subunit DnaN. To date, two conflicting models have been proposed to account for movement of two sister replisomes in vivo. In the factory model, the two replisomes are physically associated and co-migrate along DNA. Alternatively, in the tracking model, the movement of the replisomes is largely independent of each other. These two models have persisted as the long-standing controversy and the precise subcellular nature of the sister replisomes remains to be elucidated.

In this study, Zhang et al. utilized single cell analyses to obtain precise DnaN dynamics in *Caulobacter*. They demonstrate that most S phase cells harbor a single DnaN-GFP focus, which occasionally splits into two foci. This splitting is facilitated by the rapid processes of chromosome segregation and DNA compaction by the condensin SMC. Moreover, they provide evidence that the translocation speeds of each sister replisome can be differentiated when encountering replication-transcription conflicts, causing a substantial delay between the two moving replisomes and resulting in DnaN-GFP focus splitting. Their data are consistent with the idea that the sister replisomes move independently as proposed in the tracking model, but physical constraints of chromosome organization lead to their colocalization, forming a factory-like configuration.

While the research is intriguing, the experiments are well conducted, and the data are presented with clarity, a notable critique arises concerning the reliance on DnaN-GFP observations as the sole basis for conclusions. As the authors acknowledge, DnaN is loaded onto the Okazaki fragments and remains on the sister-replicated chromosomes to coordinate various processes including DNA replication and repair. Consequently, it is conceivable that the observed appearance of dim fluorescent signal split from a discrete DnaN-GFP focus, reflects the accumulation of DnaN-GFP on sister DNAs, but not on replication forks. To bolster confidence in their findings, it would be beneficial for the authors to investigate whether the dynamics of DnaN-GFP parallel those of other replisome components such as SSB and HoIB. Such complementary visualizations could provide further support to the central conclusions of the study.

Detailed comments are outlined below:

L84-86: Please indicate the time point when the dim DnaN streak appeared in Figure 1AB. If it appeared at the time point indicated by the asterisk, the statement “soon after initiation” is not true. Also, the legend does not explain the asterisk and dot in Figure 1AB.

L99: How did the authors determine the stalked pole? The stalk is invisible in Figure 1A.

L100-102: Related to the above comment, I suspect that the authors might have mis-annotated the stalked cell pole. Please clarify how the authors assess the replication events at each pole.

L107-108: I found it difficult to justify the authors' statement "symmetric relative to mid-cell", because some cells appeared to bear asymmetric signals.

L139-142: The blur foci made it difficult to justify the authors' statement. Improving the visibility of the foci is very important to draw a conclusion that two replication forks reside in close proximity to each other.

L148-149: I found it difficult to follow the statement "dim DnaN foci appeared concurrently with the time of ParB migration." On the second frame in Figure 2d, the cell appeared to harbor a single DnaN focus and two discrete ParB foci, which is consistent with the idea that the appearance of the dim DnaN foci occurs after initiation of ParB migration.

L150: The boundaries of dim DnaN foci are unclear in Figure 2e and therefore I was unable to justify the authors' statement on the correlation between ParB and DnaN.

L178: The schematic wrongly deposits DnaN on the leading strand.

L232: The authors should coherently state DnaN foci, but not replisomes. As mentioned above, it is conceivable that DnaN foci not only reflect replisome but also their accumulation on the lagging strand.

L348: The replication forks are wrongly illustrated. The authors must consider sister-replicated DNAs.

L360-364: The current data do not exclude a possible mechanism for physical tethering of the two replisomes by known/unknown proteins. Although DnaN splitting is increased in mutant cells translocating parS, this could simply mean the mutant phenotype is dominant over the intact tethering mechanism.

L383-385: In *E. coli* SeqA, CrfC, MukB are proposed to facilitate colocalization of sister replication forks (Helgesen et al. 2021 Sci Rep; Ozaki et al. 2013 Cell Rep; Sunako et al. 2001 Mol Micro).

Reviewer #2 (Remarks to the Author):

The study by Zhang et al. gives interesting and well motivated new explanations for some of the heterogeneity that is observed in chromosome replication patterns observed in *C. crescentus*.

Overall I find the work convincing but would like clarifications with respect to:

Line 156 "We found that the ParB segregation time for cells

with dim DnaN signals was on average 12.6 ± 4.4 min, as opposed to 29.8 ± 8.1 min for those with a single

DnaN focus". This observation is left uncommented I do not understand what is the conclusion from this observation.

Figure 2 "Vertical lines in kymograph correspond to the timeframe of montages and arrows highlight locus replication and segregation". I do not understand what is meant by time frame of montages and if it is replication or segregation.

Fig 2ij. I do not find the plot very convincing. Is it possible to present the lack of a dim signal in some more quantitative way ?

Fig 3f,g. What is the y-scale ?

There are also two minor things:

Line 92: " offers good photostability over high repetition and long duration time-lapse imaging."
Compared to what ?

Lin 199. Should it be 2 bright foci? May be better give the fraction of two foci cells compared to WT.

Reviewer #3 (Remarks to the Author):

In the manuscript titled "The manuscript is entitled "Chromosome organization shapes replisome dynamics in *Caulobacter crescentus*", the authors follow the replisome protein DnaN throughout the cell cycle. The study is highly relevant to the field and their findings are novel as such work has not been thoroughly investigated in *Caulobacter*. Their results are quite interesting. Furthermore, given the "factory model" controversy of the past couple of decades, this study is of great value. I applaud the authors for providing a thorough and balanced introduction and interpretation of prior work on this topic.

My main concern is that the study does not follow only one replisome protein. DnaN is a tool belt, likely involved in many DNA repair processes, therefore the conclusion that this is for sure representing the replisome is a bit of a presumption. Therefore, I would like to see at least in a couple of the key experiments: a replisome-specific protein such as the replicative helicase to be investigated. I strongly recommend this in order to solidify the conclusions.

Minor comments:

The authors state that “Our data in *C. crescentus* is also consistent with residual binding after segregation, although with a longer half-life (~7.83 min)”. How was this determined?

The sentence “SMC is a non-essential protein in *C. crescentus*, and both strains show normal replisome progression and division, but slightly slower cell growth” is grammatically incorrect. “SMC is a non-essential protein in *C. crescentus*” refers to one protein and “both ...” to two proteins

The authors say that “the majority of Δsmc cells ($58.2 \pm 1.9\%$) contained one bright focus [...] at early- to mid-replication time”. However, this seems significantly lower than in WT cells (it’s implied that it is 100%, since they only mention a dim focus). This seems like something that the authors need to explain. Especially since the authors state that “In contrast to Δsmc , splitting occurred in most *flip1-5* cells ($68.5 \pm 8.2\%$)”. This means 32% of cells have a single bright focus, so the Δsmc cells seem to have an intermediate phenotype.

Most importantly, the use of red and green on the same figure (e.g. fig 2) makes the manuscript difficult to interpret to a colorblind reader.

General remarks:

We thank all reviewers for their useful feedback, which helped us to improve the manuscript. Our replies are included below, in blue.

We have formatted the revised manuscript following the guideline of Nature Communications.

Reviewer #1 (Remarks to the Author):

In most bacteria with the circular chromosome, DNA replication occurs bidirectionally from the unique origin, forming two sister replication forks. The replisome assembled on the fork represents a highly dynamic behavior, as demonstrated by the subcellular localization patterns of the replisome components, including the clamp subunit DnaN. To date, two conflicting models have been proposed to account for movement of two sister replisomes *in vivo*. In the factory model, the two replisomes are physically associated and co-migrate along DNA. Alternatively, in the tracking model, the movement of the replisomes is largely independent of each other. These two models have persisted as the long-standing controversy and the precise subcellular nature of the sister replisomes remains to be elucidated.

In this study, Zhang et al. utilized single cell analyses to obtain precise DnaN dynamics in *Caulobacter*. They demonstrate that most S phase cells harbor a single DnaN-GFP focus, which occasionally splits into two foci. This splitting is facilitated by the rapid processes of chromosome segregation and DNA compaction by the condensin SMC. Moreover, they provide evidence that the translocation speeds of each sister replisome can be differentiated when encountering replication-transcription conflicts, causing a substantial delay between the two moving replisomes and resulting in DnaN-GFP focus splitting. Their data are consistent with the idea that the sister replisomes move independently as proposed in the tracking model, but physical constraints of chromosome organization lead to their colocalization, forming a factory-like configuration.

While the research is intriguing, the experiments are well conducted, and the data are presented with clarity, a notable critique arises concerning the reliance on DnaN-GFP observations as the sole basis for conclusions. As the authors acknowledge, DnaN is loaded onto the Okazaki fragments and remains on the sister-replicated chromosomes to coordinate various processes including DNA replication and repair. Consequently, it is conceivable that the observed appearance of dim fluorescent signal split from a discrete DnaN-GFP focus, reflects the accumulation of DnaN-GFP on sister DNAs, but not on replication forks. To bolster confidence in

their findings, it would be beneficial for the authors to investigate whether the dynamics of DnaN-GFP parallel those of other replisome components such as SSB and HolB. Such complementary visualizations could provide further support to the central conclusions of the study.

Reply: The concern that only DnaN was used was also raised by reviewer #3.

To further support our conclusions, we imaged several additional replisome proteins, SSB, HolB, and DnaB. We performed the following experiments in WT background strains:

- 1) Imaging SSB-sfGFP every 2 min over the cell cycle
- 2) Imaging both DnaN-sfGFP and SSB-mScarlet-I every 10 sec for 10 frames
- 3) Imaging both DnaN-sfGFP and SSB-mScarlet-I every 2 min over the cell cycle
- 4) Imaging HolB-YFP every 4 min over the cell cycle
- 5) Imaging DnaB-YFP every 4 min over the cell cycle
- 6) Imaging both HolB-YFP and SSB-mScarlet-I every 15 sec for 10 frames
- 7) Imaging both DnaB-YFP and SSB-mScarlet-I every 15 sec for 10 frames

The results are summarized in a new Figure 2 (below).

In brief, the DnaN dynamics does not entirely parallel SSB. We found SSB sometimes appeared in kymographs as a diffuse background rather than a trajectory originating from the bright replisome focus (Fig. 2a). In contrast, we sometimes observed two bright SSB foci at late-

replication stages, similar to what we found in DnaN. We also observed two bright HoIB and DnaB foci which generally well-localized SSB foci (Fig. 2b (ii)). We added a new section: “Late-splitting events revealed by other replisome components”, to describe these results, which add further evidence for our model of early- and late-splitting.

We also imaged SSB-sfGFP in the *flip1-5* background cell. We find $94.9 \pm 4.5\%$ of cells showed foci splitting at early-mid replication stages (examples shown below), compared to $8.6 \pm 3.1\%$ in WT background cells. This result is consistent with what we found in the DnaN imaging, supporting our conclusion that inter-arm alignment is a factor that indirectly links sister replisomes. We have added these data as Supplementary Figure 21 and Movie 5.

Detailed comments are outlined below:

L84-86: Please indicate the time point when the dim DnaN streak appeared in Figure 1AB. If it appeared at the time point indicated by the asterisk, the statement “soon after initiation” is not true. Also, the legend does not explain the asterisk and dot in Figure 1AB.

Reply: The asterisk or dot was meant to indicate the same timepoint in the image data (Fig. 1a) and kymograph (Fig. 1b). We have removed both markers to avoid confusion. We modified the Figure (below) to indicate the three essential timepoints: initiation, early splitting, and late splitting, highlighted by white, orange, and blue arrows respectively.

L99: How did the authors determine the stalked pole? The stalk is invisible in Figure 1A.

Reply: We determined the stalked pole based on the asymmetric division of *Caulobacter* cells. In a pre-divisional cell, the future stalked daughter cell is larger than the future swarmer daughter cell. For instance, in Figure 1A, we can clearly see a shape asymmetry from the last frames, thus we determine the stalked pole is in the top right corner of the image (see below). We also added a comment on stalked pole determination in the Methods section.

L100-102: Related to the above comment, I suspect that the authors might have mis-annotated the stalked cell pole. Please clarify how the authors assess the replication events at each pole.

Reply: As mentioned above, the stalked pole was determined by shape asymmetry in a pre-divisional cell if the stalk was invisible. The *ori* was then annotated at the stalked or swarmer pole at the beginning of replication.

In Supplementary Fig. 2, we show two unusual cells where replication initiates from the swarmer pole, and their kymographs. Notably, such mis-location of *ori* was only found in few cells (5.2%).

L107-108: I found it difficult to justify the authors' statement "symmetric relative to mid-cell", because some cells appeared to bear asymmetric signals.

Reply: We have decomposed the data in Figure 1d to highlight early- and late-splitting events, in which two peaks exist in the intensity profile. Individual intensity profiles were added in Supplementary Fig. 3, 4. Symmetric or asymmetric positions of the two peaks relative to mid-cell are marked by red or blue dots, respectively.

Examples of **Supplementary Fig. 3** (30% replication time) are shown below:

Examples of **Supplementary Fig. 4** (90% replication time) are shown below:

In early-splitting events, most cells (68.5%) have symmetric positions of the two foci relative to mid-cell, while in the late-splitting events it is even higher (89.7%).

Therefore, we modified the text as:

“In both cases, the positions of two foci appear symmetric relative to mid-cell in the majority of cells (Supplementary Fig. 3, 4).”

L139-142: The blur foci made it difficult to justify the authors’ statement. Improving the visibility of the foci is very important to draw a conclusion that two replication forks reside in close proximity to each other.

Reply: We agree that the foci shown in previous Fig. 2b are blurred. We imaged those cells using higher illumination power or longer exposure times but failed to achieve better signal-to-noise ratios. The potential reason could be that the low copy number of ParB-GFP (-mCherry) molecules on *parS* site, combined with a cytoplasmic background, limits the SNR.

We realized that loci splitting events are not the only features in the kymograph that should distinguish the two models. If the dim DnaN represents a functional replisome, L1 and R1 loci should be at different cell halves prior to their duplication (new Fig. 3b, below). We zoomed into the timeframes before loci duplication events in two example kymographs (new Fig. 3c, below).

Both examples showed similar trajectories for L1 and R1 (highlighted in boxes). And we found such phenomenon occurred in $96.2 \pm 4.2\%$ cells. We believe the revised figure is self-explanatory, and we revised the text as:

“...Since bright and dim DnaN foci were located at opposite cell halves at the early-replication stage, we expected to observe L1 and R1 moving to different cell halves prior to their duplications if each focus corresponded to a functional replisome (Fig. 3b). However, time-lapse montages and kymographs generally showed the trajectories of left- and right-arm ori-proximal loci residing at the same cell half until loci-splitting events occurred (Fig. 3c and Supplementary Fig. 14)...”

L148-149: I found it difficult to follow the statement “dim DnaN foci appeared concurrently with the time of ParB migration.” On the second frame in Figure 2d, the cell appeared to harbor a single DnaN focus and two discrete ParB foci, which is consistent with the idea that the appearance of the dim DnaN foci occurs after initiation of ParB migration.

Reply: We have modified the text as below:

“...dim DnaN foci appeared after the initiation of ParB migration”.

L150: The boundaries of dim DnaN foci are unclear in Figure 2e and therefore I was unable to justify the authors’ statement on the correlation between ParB and DnaN.

Reply: To better clarify, we removed triangle markers in DnaN kymographs (now Fig. 3f) so that dim signals are more clearly visible. We also realized that the word “boundary” here was confusing because we wanted to refer to the appearance of a signal in the kymograph (in time), rather than to the boundary of the dim focus (in space). We have rephrased the description in the main text as:

“... We also observed that the timing of ParB segregation matched the appearance of dim DnaN signals in kymographs...”

L178: The schematic wrongly deposits DnaN on the leading strand.

Reply: Yes, during DNA replication, the two lagging strands at two forks should segregate towards opposite cell poles (below left), which is the same for DnaN molecules on the lagging strands. We corrected the previous Figure 2c (below right).

L232: The authors should coherently state DnaN foci, but not replisomes. As mentioned above, it is conceivable that DnaN foci not only reflect replisome but also their accumulation on the lagging strand.

Reply: we have corrected that, and now state all foci corresponding to DnaN as “DnaN foci” rather than the “replisome foci”.

L348: The replication forks are wrongly illustrated. The authors must consider sister-replicated DNAs.

Reply: we have added replication forks and sister-replicated DNAs in the new figure as below.

L360-364: The current data do not exclude a possible mechanism for physical tethering of the two replisomes by known/unknown proteins. Although DnaN splitting is increased in mutant cells translocating *parS*, this could simply mean the mutant phenotype is dominant over the intact tethering mechanism.

Reply: we agree with reviewer #1. We added the following text in the discussion part:

It is still possible that unidentified proteins in *C. crescentus* could physically tether sister replisomes, but whose impact could be overcome by deformed chromosome structures (i.e. when SMC is knocked out or the *parS* site is translocated).

L383-385: In *E. coli* SeqA, CrfC, MukB are proposed to facilitate colocalization of sister replication forks (Helgesen et al. 2021 Sci Rep; Ozaki et al. 2013 Cell Rep; Sunako et al. 2001 Mol Micro).

Reply: We include these references in the revised manuscript.

Reviewer #2 (Remarks to the Author):

The study by Zhang et al. gives interesting and well motivated new explanations for some of the heterogeneity that is observed in chromosome replication patterns observed in *C. crescentus*.

Overall I find the work convincing but would like clarifications with respect to:

Line 156 “We found that the ParB segregation time for cells with dim DnaN signals was on average 12.6 ± 4.4 min, as opposed to 29.8 ± 8.1 min for those with a single DnaN focus”. This observation is left uncommented I do not understand what is the conclusion from this observation.

Reply: We revised the text as below:

“...We found that the ParB segregation time for cells with dim DnaN signals was on average 12.6 ± 4.4 min, as opposed to 29.8 ± 8.1 min for those with a single DnaN focus. Therefore, dim DnaN signals could correspond to residual replisome molecules bound to DNA behind the replication fork, which haven't yet fallen off during rapid segregation. They may appear as a streak when the DNA they are attached to is extended, or as more punctate as the DNA compacts. The

lack of dim signals may occur when the segregation time is much longer than the DnaN dissociation time, allowing time for full unloading from the segregating DNA. ...”

Figure 2 “Vertical lines in kymograph correspond to the timeframe of montages and arrows highlight locus replication and segregation”. I do not understand what is meant by time frame of montages and if it is replication or segregation.

Reply: The vertical lines in kymographs (previous Fig. 2b, below left) were used to indicate a range of times, corresponding to the raw images that were presented as montages (previous Fig. 2b, below right). These raw images were chosen because splitting from one to two loci appeared, indicating the labeled genomic locus was replicated and segregation.

We modified the previous Fig. 2a-c (now Fig. 3a-c) because we did not find the montages to be helpful for interpreting the data. However, we used the same annotation in the new Fig. 2 (example shown below).

We now have rephrased the text in figure caption as below:

“Vertical lines in kymographs correspond to specific time ranges, during which individual images are shown as montages.”

Fig 2ij. I do not find the plot very convincing. Is it possible to present the lack of a dim signal in some more quantitative way ?

Reply: We quantified the differences between strains by comparing the number of DnaN foci detected during replication in WT and *parAK20R* background cells. We found that only 0.8% of

parAK20R mutant cells contained two DnaN foci, which is fewer compared to 7.2% in the WT case (below).

We added the figure above to the supplementary information, and refer to it in the main text as below:

“We also detected DnaN foci in WT and *parAK20R* background cells, and found that only 0.8% of *parAK20R* cells contained two foci (with symmetric or asymmetric intensities), which is reduced ~10 times compared to the WT (Supplementary Fig. 18).”

Fig 3f,g. What is the y-scale ?

Reply: There is no y-scale in this figure. It is a stack of three horizontal violin plots, corresponding to the three different strains we imaged.

There are also two minor things:

Line 92: “ offers good photostability over high repetition and long duration time-lapse imaging.”
Compared to what ?

Reply: We did not quantitatively compare the photostability between different fluorescent proteins in this study. We replaced the word “good” into “sufficient” in the revised manuscript.

Lin 199. Should it be 2 bright foci? Maybe better give the fraction of two foci cells compared to WT.

Reply: Our description here was confusing, also pointed out by reviewer #3.

In contrast to commenting on the cells containing “one bright focus”, we now comment on the fraction of cells containing “two bright foci”, and rephrased the sentence as:

“Kymographs of DnaN fluorescence revealed that many Δsmc cells ($41.8 \pm 1.9\%$) contained two bright foci at early- to mid-replication time (Fig. 4b, Supplementary Fig. 19 and Movie 3), which was rarely observed in WT cells ($6.9 \pm 3.9\%$) (Supplementary Fig. 1)”

Reviewer #3 (Remarks to the Author):

In the manuscript titled "Chromosome organization shapes replisome dynamics in *Caulobacter crescentus*", the authors follow the replisome protein DnaN throughout the cell cycle. The study is highly relevant to the field and their findings are novel as such work has not been thoroughly investigated in *Caulobacter*. Their results are quite interesting. Furthermore, given the “factory model” controversy of the past couple of decades, this study is of great value. I applaud the authors for providing a thorough and balanced introduction and interpretation of prior work on this topic.

My main concern is that the study does not follow only one replisome protein. DnaN is a tool belt, likely involved in many DNA repair processes, therefore the conclusion that this is for sure representing the replisome is a bit of a presumption. Therefore, I would like to see at least in a couple of the key experiments: a replisome-specific protein such as the replicative helicase to be investigated. I strongly recommend this in order to solidify the conclusions.

Reply: A similar concern was raised by reviewer #1, main comment 1. We now have done more experiments on additional replisome proteins and added new data to the revised manuscript. Please find our detailed responses above.

Minor comments:

The authors state that “Our data in *C. crescentus* is also consistent with residual binding after segregation, although with a longer half-life (~ 7.83 min)”. How was this determined?

Reply: We estimated the half-life by modeling the unloading of DNA-bound β -clamp molecules as an exponential decay. However, we did not perform single-molecule tracking experiments as others have for measuring the half-life time (e.g. Moolman et al., 2014, Nat Commun; Beattie et

al., 2017, eLife), so our estimate is not precise. For simplicity, we removed the estimation in our revised manuscript:

“..Notably, the dissociation time of per DnaN clamp (β 2 dimer) measured in living *E. coli* cells can be up to minutes long (0.78 or 2.75 min on average), where more than half of the molecules in the cell were bound to DNA...”

The sentence “SMC is a non-essential protein in *C. crescentus*, and both strains show normal replisome progression and division, but slightly slower cell growth” is grammatically incorrect. “SMC is a non-essential protein in *C. crescentus*” refers to one protein and “both ...” to two proteins.

Reply: We have corrected the sentence as: “... SMC is a non-essential protein in *C. crescentus*, and the Δsmc strain shows normal replisome progression and division...”.

The authors say that “the majority of Δsmc cells ($58.2 \pm 1.9\%$) contained one bright focus [...] at early- to mid-replication time”. However, this seems significantly lower than in WT cells (it’s implied that it is 100%, since they only mention a dim focus). This seems like something that the authors need to explain. Especially since the authors state that “In contrast to Δsmc , splitting occurred in most *flip1-5* cells ($68.5 \pm 8.2\%$)”. This means 32% of cells have a single bright focus, so the Δsmc cells seem to have an intermediate phenotype.

Reply: A similar comment was made by reviewer #2. We rephrased as below:

“Kymographs of DnaN fluorescence revealed that many Δsmc cells ($41.8 \pm 1.9\%$) contained two bright foci at early- to mid-replication time (Fig. 4b, Supplementary Fig. 19, and Movie 3), which was rarely observed in WT cells ($6.9 \pm 3.9\%$) (Supplementary Fig. 1).”

“...We also imaged SSB-sfGFP in *flip1-5* background cells every 2 min, and found such splitting appeared in most cells ($94.9 \pm 4.5\%$) at early-mid replication stages (Supplementary Fig. 21 and Movie 5). Overall, the *smc* knockout results in an intermediate phenotype between WT and *flip1-5* for the foci-splitting events”

Most importantly, the use of red and green on the same figure (e.g. fig 2) makes the manuscript difficult to interpret to a colorblind reader.

Reply: We have modified figures to avoid using red and green colors in the same figure.

REVIEWERS' COMMENTS

Reviewer #1 (Remarks to the Author):

The authors appropriately responded to my concerns and accordingly the current version well supports the conclusion of this study.

Minor comment:

Please double-check the strain CB15N :: dnaB-sfGFP :: P_{xyl}-SSB-mScarlet-I. Should it be CB15N :: dnaB-YFP :: P_{xyl}-SSB-mScarlet-I?

Reviewer #2 (Remarks to the Author):

This is an important paper that I believe should be published.

The authors have made a good revision.

The Zenodo link: <https://doi.org/10.5281/zenodo.10203421> does not work for me.

I have no further comments.